# Hydrocarbon Sorption in Flexible MOFs—Part II: Understanding Adsorption Kinetics

**DOI:** 10.3390/nano13030601

**Published:** 2023-02-02

**Authors:** Hannes Preißler-Kurzhöfer, Andrei Kolesnikov, Marcus Lange, Jens Möllmer, Oliver Erhart, Merten Kobalz, Seungtaik Hwang, Christian Chmelik, Harald Krautscheid, Roger Gläser

**Affiliations:** 1Institut für Technische Chemie, Fakultät für Chemie und Mineralogie, Universität Leipzig, Linnéstraße 3, D-04103 Leipzig, Germany; 2Institut für Nichtklassische Chemie e.V., Universität Leipzig, Permoserstraße 15, D-04318 Leipzig, Germany; 3Institut für Anorganische Chemie, Fakultät für Chemie und Mineralogie, Universität Leipzig, Johannisallee 21, D-04103 Leipzig, Germany; 4Fakultät für Physik und Geowissenschaften, Universität Leipzig, Linnéstraße 5, D-04103 Leipzig, Germany

**Keywords:** metal–organic frameworks, kinetic analysis, flexible materials

## Abstract

The rate of sorption of *n*-butane on the structurally flexible metal-organic framework [Cu_2_(H-Me-trz-ia)_2_], including its complete structural transition between a narrow-pore phase and a large-pore phase, was studied by sorption gravimetry, IR spectroscopy, and powder X-ray diffraction at close to ambient temperature (283, 298, and 313 K). The uptake curves reveal complex interactions of adsorption on the outer surface of MOF particles, structural transition, of which the overall rate depends on several factors, including pressure step, temperature, as well as particle size, and the subsequent diffusion into newly opened pores. With the aid of a kinetic model based on the linear driving force (LDF) approach, both rates of diffusion and structural transition were studied independently of each other. It is shown that temperature and applied pressure steps have a strong effect on the rate of structural transition and thus, the overall velocity of gas uptake. For pressure steps close to the upper boundary of the gate-opening, the rate of structural transition is drastically reduced. This feature enables a fine-tuning of the overall velocity of sorption, which can even turn into anti-Arrhenius behavior.

## 1. Introduction

Flexible MOFs, or PCPs of the third generation as described by Kitagawa, have been studied extensively since the early 2000s [1,2]. Due to their tuneable pore geometry, rich configurational diversity, and unique structural dynamics upon adsorption [3,4,5,6,7], these materials have become promising candidates as adsorbents in important industrial fields like gas separation [8,9,10] and storage [11,12], in sensor design [13,14] or in drug delivery systems [15,16]. To drive this exciting class of materials towards applicability, information about the sorption capacities in the given phases, the gate-opening conditions and their thermal, mechanical, and chemical stabilities, in addition to gas uptake kinetics and the subsequent heat transfer, must be garnered. During the last decade, much effort has been invested into recording adsorption equilibrium data for known flexible MOF families, such as MIL [17,18,19], DUT [7,20,21] or [Zn(bdc)(dabco)] [3,22,23,24], while also studying these materials with respect to their decomposition under hot or wet conditions [25,26]. However, uptake kinetics have been studied relatively scarcely in the field of flexible MOFs. 

The few reports on the sorption kinetics of flexible materials so far are mostly concerned with small pressure steps within the gate-opening of these materials at cryogenic temperatures, mostly using noble gases, O_2_ or N_2_. The majority of these uptake curves have in common that an inflection point occurs, indicating the superposition of at least two processes during the uptake. Furthermore, equilibration times are much more extended compared to more conventional uptake curves on rigid materials like zeolites or carbon. In order to describe this idiosyncratical adsorption uptake, mainly semi-empirical models are used. For example, the double exponential (DE) model based on a linear driving force (LDF approach), taking into account two separate diffusion processes, found vast application in various reports over the years while no direct connection to the transition rate is possible [27,28,29]. Furthermore, Lueking et al. [30] investigated both LDF-exceeding stretched and compressed exponential models (SE/CE-models), which require an additional exponential factor to the LDF-equation, in order to fit uptake curves of N_2_, O_2_, and argon on the flexible MOF RPM3-Zn at 77 K and 87 K. The fit of the curves had a good overlap with the experimental data, with the interpretation being based on the Avrami theorem regarding nucleus growth, which makes the comparison to the real-world system quite challenging. A more sophisticated model was developed by Tanaka et al. [31]. Known as the “GO-Model” (gate-opening-model), a kinetic constant for the evolution of the gate-opening (*k_GO_*) is introduced in addition to two rate constants, describing diffusion in two different pore channels within the expanded MOF [Cd(bpndc)] upon adsorption of N_2_ and O_2_ in a temperature range of 77 to 90 K. 

While the bare kinetic behavior of flexible MOFs upon hydrocarbon adsorption at ambient temperatures was not yet reported to the best of our knowledge, various reports deal with the potential separation of hydrocarbon gas mixtures [8,9,32,33,34] using this class of materials. Therefore, this work focuses on the investigation of the interplay between diffusion and the structural transition in a flexible MOF described by a mathematical model for gas uptake at various temperatures. Exemplary for a hydrocarbon/flexible MOF system, *n*-butane as a probe molecule on [Cu_2_(H-Me-trz-Ia)_2_] (**1**) was studied, which was previously investigated thermodynamically [35]. Gravimetric uptake and IR diffusion measurements were conducted at different temperatures and for different pressure steps according to given adsorption isotherms. In addition, the effect of particle size distribution on the adsorption kinetics has been studied as well as the influence of several adsorption-desorption cycles on the particle size distribution.

## 2. Materials and Methods

### 2.1. Materials

The MOF [Cu_2_(H-Me-trz-Ia)_2_] (**1**) studied herein is part of an isostructural series of metal-organic frameworks built on Cu^2+^ metal ions and triazolyl-isophthalate linkers [36]. The adsorbent was further presented in a recent publication regarding the thermodynamic analysis of the gate-opening using the Dubinin-based universal adsorption theory [35]. It was shown that the material switches from a narrow-pore form (*np*-form) to a medium pore form (*mp*-form) upon adsorption with *n*-butane while CO_2_ can even trigger a second step towards a large-pore form (*lp*-form). Structurally, the bridging coordination of the carboxylate groups of the linkers results in a square planar CuO_4_ environment, leading to the well-known dinuclear paddle wheel motif. Through the coordination of nitrogen atoms of the triazolyl groups to the metal centers in the apical positions, a three-dimensional network is assembled.

The synthesis of [Cu_2_(H-Me-trz-Ia)_2_] (**1**) was conducted according to the original procedure reported [36]. The crystal structure data, as well as further illustrations, are given in the Appendix A.

### 2.2. Methods

#### 2.2.1. Sorption Isotherms

Isotherms were measured according to a modified protocol by Keller and Staudt [27]. The adsorption and desorption isotherms of ethane, propane, and *n*-butane on the MOFs were determined in a temperature range from 283 to 313 K and at pressures of up to 5 MPa using a magnetic suspension balance (Fa. Rubotherm GmbH, Bochum, Germany). Three pressure transducers (MKS Instruments Deutschland GmbH, Munich, Germany, Newport Omega Electronics GmbH, Deckenpfronn, Germany) were used to collect data for the pressure range up to 5 MPa. Before the sorption experiments, the MOFs (0.2 g) were activated for at least 12 h at 373 K under a minimum pressure of 0.3 Pa until constant mass was achieved. Materials were used only for a maximum number of 10 cycles, which avoids any cycling stability issues with this series of MOFs. The temperature was kept constant throughout the measurement with an accuracy of 0.5 K. Ethane, propane, and *n*-butane were obtained from Linde (Linde AG, München, Germany) with purities of 99.5%. All isotherms within this work are presented in absolute gas loading based on a buoyancy correction [27].

#### 2.2.2. Sorption Kinetics Investigation—Gravimetric Measurements

The gravimetric sorption kinetics measurements were conducted using a magnetic suspension balance (Fa. Rubotherm GmbH, Bochum, Germany) according to Möller et al. [37]. To minimize bed diffusion effects within the sample and to ensure good heat transport between the sample and its surroundings, 400 mg of **1** were thinly distributed on a special sample holder (see Appendix A). The sample cell was evacuated for at least 12 h at 373 K and 0.3 Pa until constant mass was achieved. To ensure constant pressures, an additional gas reservoir of 5000 cm^3^ equipped with a pressure transducer was used. The probe molecule n-butane was filled in the gas reservoir, and the valve connected to the sample cell was completely opened after achieving constant temperature and pressure conditions. Sorption equilibrium was assumed to be reached when no further weight increase within 15 min was observed. 

#### 2.2.3. Sorption Kinetics Investigation—IR-Measurements

The methodology of IR microscopy for sorption and diffusion measurements has been described in detail elsewhere [38,39] and is only briefly summarized here. The procedure is based on monitoring the intensity of characteristic IR bands of the guest molecules, which is known to be proportional to their concentration according to the Lambert–Beer law. The setup consists of a Bruker HYPERION 3000 IR microscope which is attached to a Bruker VERTEX 80v FTIR spectrometer (Bruker Optics GmBH, Ettlingen, Germany). Small amounts of **1** soaked with MeOH were introduced into the IR cell. For activation, the IR cell containing the sample was heated to 328 K with a heating rate of 5 K min^−1^ and kept under vacuum at this elevated temperature for 12 h. For the measurement, the IR cell is mounted on the movable x-y sample stage of the microscope. After the selection of an individual area of particle agglomerates of the sample in visible mode, all IR measurements were performed in IR transmission mode at 298 and 313 K, respectively. For the screening of transport diffusivities $D_{T}$ in dependence of the amount adsorbed, each uptake curve was fitted with a simple LDF model and the diffusivity approximated with the formula $D_{T}=\frac{k\;d^{2}}{15}$ [40], where *k* is the fitting constant and *d* is the crystal diameter. 

#### 2.2.4. Sorption Kinetics Investigation—In-Situ PXRD-Measurements

A straightforward apparatus was built for pressure-dependent powder X-ray diffraction [41]. A gas supply was connected to a sample capillar mounted on a commercial X-ray diffractometer in Debye–Scherrer geometry (StadiP, STOE and Cie. GmbH, Darmstadt, Germany) equipped with a sealed X-ray tube (Cu-K_α1_ radiation, λ = 154.060 pm) and a DECTRIS Mythen detector (DECTRIS AG, Baden-Daettwil, Switzerland). A microcrystalline sample of **1** was filled into a glass capillary (outer diameter 1 mm) as the measuring cell and activated by connecting to a vacuum pump before each measurement. The adsorptive gas was admitted from a reservoir to the measuring cell through a dosing valve. The measurements were carried out at room temperature by applying 10 kPa of *n*-butane to the evacuated sample. The pressure in the cell was monitored over time using a pressure transducer (Newport Electronics GmbH, Germany). Diffraction patterns were recorded every 2.6 s, and the experiment was terminated after 10 min without apparent change in the structure during the last PXRD scans. In order to calculate the state of transition in dependence on time, the reflection intensities at around 2θ = 13° were integrated and normalized (see Appendix A). 

#### 2.2.5. Characterization of MOF-Particles—SEM Images

SEM imaging was performed on an LEO 1530 (LEO Electron Microscopy Ldt., Cambridge, UK) under vacuum conditions with an acceleration voltage of 20 kV, detecting secondary and backscattered electrons. EDX mapping was performed with an INCA-analysis system (ETAS GmbH, Stuttgart, Germany). Prior to the imaging, the samples were carbon vapor treated with a CED 030 from Balzers (Oerlikon Balzers, Balzers, Liechtenstein). 

#### 2.2.6. Characterisation of MOF-Particles—Particle Size Distribution Measurements

Particle size distribution measurements were conducted on a Cilas 1064 (Quantachrome GmbH & Co. KG, Odelzhausen, Germany). The diffraction angles of a wet dispersion of **1** in water were analyzed using a monochromic laser beam (λ = 1064 nm). Calculations were based on spherical particle shapes. 

#### 2.2.7. Kinetic Fitting Model Description

The model used within the main manuscript is based on the GO model of Tanaka et al. [31] with slight alterations. It does utilize the linear driving force model (LDF model) and is specifically useful to describe the beginning of gas uptakes in flexible materials for pressure jumps close to the point of the structural transition. In the model approach presented here, the overall velocity of gas uptake is separated into three distinct processes, as stated in Section 3.3. The (i) diffusion on the outer particle surface, the (ii) gate-opening, and the (iii) diffusion into newly opened pores. Within Tanaka’s model, the rate constant for the gate-opening $k_{GO}$ was coupled with two diffusion constants for the diffusion within two separate pore geometries. However, within the system at study herein, there is thought to be only one pore size present, and thus, only one rate constant was implemented for the diffusion into newly opened pores $k_{D2}$. One rate constant of adsorption was added for the surface adsorption $k_{D1}$, which starts the process but is not coupled to the rate constant of the structural transition $k_{GO}$. Therefore, the whole gas uptake in dependence on time can be described by Equation (1).
(1)$$F\left(t\right)=A\left\{1-\exp\left(-k_{D1}t\right)\right\}+\left(1-A\right)\left\{1-\frac{k_{GO}\exp\left(-k_{D2}t\right)-k_{D2}\exp\left(-k_{GO}t\right)}{k_{GO}-k_{D2}}\right\}$$

In this equation, A is a supplementary parameter in addition to the three rate constants and marks the limit to which fractional loading on the surface adsorption is correlated. Thus, A should adopt a value between 0 and 1. Within this work, the value of A was fixed to 0.05 based on the sorption isotherms, reducing the model essentially to the three rate constants as fitting parameters. 

The fitting approach was processed within EXCEL and individual mathematical solvers to minimize the residuals. Herein, all parameters were provided boundaries in order to find reliable solutions. Furthermore, the focus was placed on the first section of the uptake in order to get more insights into the inflection points and structural transition, while larger residuals were allowed within the longer equilibration times, which are governed by other effects. These are further idiosyncrasies (shown in the Appendix A) in gas uptakes within flexible materials that are not represented by Equation (1). These will be tackled in future works.

## 3. Results

### 3.1. Sorption Isotherms of n-Butane on [Cu_2_(H-Me-trz-Ia)_2_] (**1**)

The sorption isotherms of *n*-butane on **1** at 283, 298, 313, and 328 K are shown in Figure 1. The stepped, sigmoidal shapes in the surface excess vs. log p diagrams typical for flexible materials are displayed for various temperatures (Figure 1 left) and with the use of the potential theory in a traditional Dubinin plot, where all isotherms are superimposed to one characteristic pattern (Figure 1 right). From the latter, a temperature-independent dual Dubinin–Asthakov fit can be derived with only minor deviations for the underlying system (see Appendix A). With this modeling, the boundaries of the structural transition or so-called gate-opening, herein referred to as gate-opening start (*GOS*) and gate-opening end (*GOE*), respectively, can be mathematically approximated using the excess surface work theory [42] (see Appendix A). The precise thermodynamics governing the sorption process of the studied system, as well as the analytical framework utilizing the Dubinin theory, were investigated in a previous work [35]. The basic model conception of adsorption is summarized in the following.

Generally, the uptake in the low-pressure region up until the *GOS* is small, evidenced by a loading of 0.05–0.16 mole gas per mole unit cell (mol mol^−1^) for temperatures between 283 and 328 K. This can be interpreted as the adsorption of *n*-butane on the outer surface area and within the spatial of the pore entries on the *np*-form of the framework. However, as soon as a specific pressure is reached, the *mp*-form becomes energetically preferred [35]. Triggered by the adsorption stress that is exerted by the adsorptives within the pore entries [5], the structure is destabilized, pressed open, and thus, will start to switch into the larger pore form of the MOF—in a domain-by-domain mode [43]—and generate a microporous pore structure accessible for gas molecules. Due to different sizes of MOF particles and the analog distribution of energy needed to open the domains of the framework, as well as a kinetic hindrance that is assigned to the adsorption process specifically, the phase transition happens in a pressure range rather than at a sharp pressure point [35]. Once the structural transition is complete, further pressure increase leads to minor pore filling of the *mp*-form of **1** up to saturation loadings between 3.7 and 3.4 mol mol^−1^ in the investigated temperature range. Thus, there is only little adsorption outside of the gate-opening range, indicating that the structural transition leads to the largest loading increase on the material and plays, therefore, a pivotal role during the overall sorption kinetics. 

### 3.2. Structural Transition and the Effect on the Particle Size

By using a self-built in-situ powder X-ray diffraction (PXRD) sample holder [41], it was possible to monitor the structural changes by means of changes in the diffraction patterns during the ad- and desorption of *n*-butane on **1**. Appendix A shows equilibrium points in the isotherm and their corresponding X-ray diffraction patterns. In order to monitor this structural transition under a rapid pressure increase, the kinetic uptake of *n*-butane on **1** was registered via PXRD at room temperature. A glass capillary was filled with the MOF sample, and the X-ray primary beam was focused on the upper part of the powder sample (see Appendix A) to avoid diffusion limitations. After the prompt application of 10 kPa of *n*-butane at 298 K, a pressure sufficient to open the framework, a rapid transition of the framework was observed, as shown in Figure 2. The powder diffraction patterns at 0 s and 200 s match the patterns of a fully evacuated *np*-form and the opened *mp*-form derived from the equilibrium measurements, respectively. The transformation becomes most obvious for the reflections around 2θ = 11°, where a growing reflection can be assigned to the *mp*-structure, and around 2θ = 13°, where a shrinking reflection can be assigned to the *np*-form. 

Interestingly, this experiment was only reproducible after the third time it was conducted using the same sample (see Appendix A). During the first two tries, reflections of the *mp*-form appeared not until several minutes after the gas application. It became furthermore obvious that the signal intensity decreased from the first turn to the third, indicating decreasing particle sizes. This is consistent with the results of Mason et al., where the particle sizes in flexible frameworks decrease upon the internal stress of several adsorption-desorption cycles [12]. SEM images of a freshly synthesized, evacuated sample and a sample that witnessed twenty adsorption-desorption cycles show that the particle size decreases upon adsorption with a great extent of fractures on the surface for the latter (Appendix A). The particle size distributions for different numbers of adsorption-desorption cycles are presented in Figure 3. Already after one cycle, smaller particle sizes are observed. After five and after twenty cycles, the particle size distributions show medians of 2.9 and 3.1 µm, respectively, indicating a stable particle size after at least five cycles. As explained above, the adsorption on the surface of the *np*-form of **1** is crucial to the gate-opening, since it is likely that *n*-butane cannot enter the framework in this state. With decreasing particle size, the overall accessible surface area for these particles is increasing, meaning available adsorption sites increase as well. This leads to a higher probability of a gate-opening as observed during the XRD-monitored structural transition. Therefore, the investigated uptake experiments shown below were all conducted after the sample experienced at least five adsorption-desorption cycles, so an equal, stable, and reproducible particle size distribution can be assumed.

### 3.3. Characteristics of Sorption Kinetics in Flexible MOFs

In order to study the kinetics of the gate-opening of **1** triggered by *n*-butane in detail, two additional techniques were applied. While IR-monitored sorption experiments are suitable for small pressure steps and the subsequent evaluation of the diffusivities [44], gravimetric measurements are robust enough to study gas uptakes to much higher pressures [37]. In the first investigation, the transport diffusivity coefficients were screened in dependence on the loading in the MOF using the IR technique. From Figure 4, it becomes obvious that there exist at least three distinct diffusion regimes. It has to be stated that the majority of uptakes at relatively low and high pressures are completed almost instantaneously, leading to the conclusion that the diffusion is too fast to be observable under the used resolution. Thus, within the bare *np-* and *mp*-forms, the transport diffusivity is at least around one order of magnitude larger compared to the mixed regime during the structural transition. Herein, molecules diffusing through already opened cell domains are hindered at closed domains until these are opened, too. Thus, the overall velocity of gas uptake decreases, but this is rather correlated to the triggering of the structural transition than to the bare mobility of the guest molecules inside the defined pore system. This apparent transport diffusivity already indicates that rather the gate-opening is the rate-determining step of the investigated process than the bare diffusion process.

The gravimetric uptake after a pressure step from 0 to 10 kPa at 298 K—the same settings as those used during the PXRD investigation—is shown in Figure 5. The shape of the uptake curve reveals two characteristic and significant deviations from diffusion behavior as generally described by Fick’s law. First, an inflection point after around 10 s and a smooth increase in loading even after a long equilibration time (for further representations see Appendix A). 

Firstly, the inflection point can be explained as the first adsorption step on the outer particle surface, which is then followed by a much slower structural transition. Secondly, an observed long-term behavior is usually accounted to non-isothermal behavior. An increased temperature due to sorption heat temporarily reduces the amount adsorbed and the subsequent cool-down to the initial temperature leads to a prolonged adsorption process. However, during IR uptakes, the same prolonged development of mass increase was observed as well. The herein used very small sample mass and the large sample holder relative to that would not enable such drastic heat effects [39], indicating that there is another factor governing the long-term behavior of sorption uptakes in the studied system. The distribution of particle sizes, as seen in Figure 3, infers a distribution of energy needed for the structural transition [45], which may furthermore affect the rate of structural transition. This is further indicated by the in-situ PXRD experiment itself, in which the first tries with larger crystal sizes took much more time (see Appendix A). The XRD intensities from Figure 2 were furthermore integrated, normalized, and plotted as a relative uptake in Figure 5 as well. These states of structural transition in dependence on time are not in complete agreement with the gravimetric uptake, however, they show the same characteristic shape and long-term behavior (see also Appendix A). This further strengthens the argument that the gate-opening mostly governs the overall velocity of gas uptake.

These findings lead to the following assumptions: First, a surface diffusion and adsorption process on the *np*-form takes place as it is unlikely that adsorptives can enter the structured pore system due to a narrow pore width. Secondly, adsorbed gases at the pore entries of the *np*-phase can trigger a structural transition. This process itself is particle-size-dependent as bigger particles have a slower rate for the gate-opening compared to smaller particles and thus long equilibration times are observed. Diffusion and adsorption into newly opened pores then further trigger structural transitions in still closed crystal domains, leading to a coupled mechanism of diffusion and gate-opening. 

Thus, the proposed model in Section 2.2.7 was built on these assumptions. The first diffusion on the outer surface is described by the rate constant $k_{D1}$, enabling the modeling of the inflection point at the beginning of the sorption uptake. The overall uptake in this part is limited to an overall fraction of 10% of the whole uptake, taking into account the limited loading in the isotherm at low pressures in this phase. This is followed by the model proposed by Tanaka et al. [31], taking into account one rate constant for the diffusion into newly opened crystallographic domains $k_{D2}$, as well as one rate constant for the structural transition $k_{GO}$. More generally, $k_{GO}$ incorporates all processes not included by $k_{D1}$ and $k_{D2}$, which includes the hindered diffusion into pore entries of the *np*-phase and the adsorption within there. Furthermore, in order to trigger a structural transition within one crystallographic domain, several adsorbed molecules may have to be present within said pore entries to induce enough mechanical stress. Thus, $k_{GO}$ describes the time dependency of the entirety of the processes involved for the structural transition to occur for all crystallographic domains in all particles of the probed sample. The model is thus reduced to three parameters, given a fixed fraction for the first phase for all uptakes. The overall schematics of the whole process are represented in Figure 6. 

The derived fit shown in Figure 5 is based on this model and is constructed for an isothermal case. As can be seen, the previously stated long-term behavior due to particle size dependency of $k_{GO}$ is not taken into account as this would (a) drastically increase the complexity of the model and (b) usually the beginning of gas uptakes are of higher importance as compared to the long-term behavior. However, further model varieties regarding non-isothermal behavior as well as particle size dependency on the rate of sorption are currently being developed and will be published in the future.

### 3.4. Temperature Dependence of the Sorption Kinetics

Since the gate-opening start and end pressures ($p_{GOS}$ and $p_{GOE}$) are temperature-dependent, the pressure step required to completely open the framework increases with increasing temperature. Thus, it is not trivial to identify experimental conditions that allow a fair comparison of uptakes at different temperatures. In the first investigation, the pressure was varied from 0 to *p*/$p_{0}$ = 0.04 of *n*-butane. The corresponding pressure steps were 0 to 6, 10, 15, and 23 kPa for 283, 298, 313, and 328 K, respectively. Figure 7 shows the resulting gas uptakes as a function of time. A remarkable feature of the uptake curves is noticeable in the temperature dependence that switches from Arrhenius behavior before ~15 s to anti-Arrhenius behavior after 15 s (especially Figure 7 right). 

Interestingly, the fastest uptake is not only at the lowest temperature (283 K) but also with the smallest absolute pressure step. Such anti-Arrhenius behavior is commonly observed for gas uptakes during gate-opening in flexible MOFs [27,28,29], and similar observations were reported for the inclusion of organic vapors in clathrate hosts [46,47,48]. 

Fitting the uptakes with the isothermal model presented in Section 3 (Equation (1)) leads to the fitting parameters $k_{D1}$, $k_{D2},$ and $k_{GO}$ for all four uptakes given in Table 1. Within the range of 0 ≤ Ψ ≤ 0.5 of relative uptake, the model fits are in good agreement with the experimental data. However, the deviation between the model and data above 0.5 relative uptake increases with increasing temperature. This would indicate that the particle size dependency of the process evolves in the same order, meaning very small dependency at 283 K (almost perfect fit) compared to a large impact on the overall adsorption process (largest deviation) at higher temperatures.

From the model parameters, it becomes obvious that both rate constants of diffusion $k_{D1}$ and $k_{D2}$ increase with increasing temperature, as would generally be expected. More precisely, the values of $k_{D1}$ and $k_{D2}$ at 298 K would translate to a transport diffusivity of around 10^−13^ m^2^ s^−1^, which is in good agreement to other transport diffusivities of *n*-butane in microporous materials [37]. Furthermore, the same order of magnitude was observed for the diffusivities during the desorption, which is not thought to be hindered by the structural transition (see Appendix A). On the contrary, the rate constant of the structural transition itself decreases with increasing temperature, leading to the described anti-Arrhenius behavior. Thus, the question arises as to why temperature and $k_{GO}$ are inversely correlated within this set of uptakes.

.

Generally, the change in chemical potential dμ is regarded as the driving force for the diffusion flux $J_{i}$ [49], and according to the Dubinin theory, it is mathematically the same as the sorption potential *A*. For the remainder of this work, it will be referred to as the sorption potential *A*.
(2)$$-d\mu =A=RT\ln\frac{p_{0}}{p_{app}}$$

Herein, R is the universal gas constant while $p_{0}$ and $p_{app}$ denote the saturation pressure and applied pressure in the gas uptake experiment, respectively. Since, in this experimental setup, the ratio $\frac{p_{0}}{p_{app}}$ remains constant, one would expect an increase in the rate constant of diffusion with increasing temperature, as is the case. However, the rate of the structural transformation for the entirety of the framework, $k_{GO}$, decreases with increasing temperature. 

The driving force for the structural transition is also a change in the applied chemical potential, which triggers the adsorption stress, but, in contrast to regular adsorption, a minimum chemical potential has to be overcome in order to trigger the gate-opening. Although the structural transition energy barrier is rather an energy distribution than a definite value, this minimum value of chemical potential is herein assumed to be $A_{GOE}$. Therefore, the applied adsorption potential must exceed the adsorption potential of the fluid at the gate-opening end (*GOE*) in order to guarantee a complete transition. This condition is fulfilled for all four gas uptakes, nevertheless, since the gate-opening pressure range becomes broader with increasing temperature, the pressure steps of 0 → 0.04 *p*
$p_{0}^{-1}$ get closer to $p_{GOE}$, the higher the applied temperature is (see Appendix A). In order to take into account the relative position of $p_{GOE}$ and to properly quantify the driving force of the structural transition, a fixed change in adsorption potential dA for all temperatures is needed in order to investigate the temperature dependence.
(3)$$dA_{GO}=A_{app}-A_{GOE}=RT\ln\frac{p_{app}}{p_{GOE}}$$

Considering the results of the first investigation (Figure 7), it has to be stated that both applied temperature and applied pressure are changed without taking into account the different changes in chemical potential, similar to reports with the same observations [29,31]. Since the boundary conditions of the structural transition are so sensitive to these two parameters, it is possible that the apparent anti-Arrhenius behavior is an artifact, occurring only under specific experimental conditions. Therefore, in order to further investigate the temperature influence, two comparative studies with constant changes in adsorption potential were conducted. For that purpose, pressures were chosen that lead to a change in adsorption potential of $dA_{GO}$ = 4100 J mol^−1^ and $dA_{GO}$ = 5700 J mol^−1^ (both are marked S1 and S2 in Figure 8, respectively. The precise pressure steps are tabulated in Appendix A). The resulting uptake curves can be seen in Figure 9.

Since these pressure steps are largely overdosing the $p_{GOE}$ ($p_{app}$ >> $p_{GOE}$), the curves attain more regular shapes, showing less pronounced inflection points and, therefore, cannot be distinguished anymore into three distinct segments. This is probably due to the fact that the energy necessary to open the framework is provided almost instantaneously after the pressure application, and thus, the rate of structural transition becomes less dependent on the particle size. All fitting parameters of the isothermal model are presented in the Appendix A. 

From these uptakes, it becomes clear that the temperature dependence does follow an Arrhenius behavior, meaning higher applied temperatures combined with larger pressure steps lead to faster uptakes. Thus, given the results of the first investigated set of uptakes (Figure 7), there has to be a crossover where identical uptake curves can be measured at a combination of lower temperature with a smaller pressure step and at higher temperature combined with a larger pressure step (see Appendix A). This is experimentally shown in Figure 10, where an uptake at 298 K and a pressure step 0 → 20 kPa overlaps with only small deviations with an uptake at 313 K and a pressure step 0 → 26 kPa. Given the isotherms at these temperatures, the differences in loading at these conditions are marginal. 

Therefore, the relative position of a pressure step in a Dubinin plot appears to be related to the rate of the structural transition $k_{GO}$ and subsequently, the velocity of sorption. This leads to the conclusion that the rate of structural transition is mainly influenced by the energy difference between the two states for the respective structures (bare *np*-form of [Cu_2_(H-Me-trz-ia)_2_] (**1**) and *mp*-form with guest molecules) at given temperature and pressure and thus is rather described by basic thermodynamic relations, such as free-energy profiles. Using the latter from molecular simulations, the transition state theory (TST) can be used to calculate a theoretical rate of the structural transition for a perfect crystal. This was done already by Numaguchi et al. for a model system [45], but no application of the TST for a real flexible system was conducted, to the best of our knowledge. However, Free Energy profiles are currently a hot topic in the computational chemistry community of flexible materials [50,51,52] and further advancements are to be expected. The possibility of approximating the rate of the gate-opening and finding optimal pairs of temperature and pressure via large-scale computational screenings could open possibilities for efficient, cost-effective sorption applications. Given the fact that gas separations of enormous industrial importance (e.g., CH_4_/N_2_ separation) most often show high thermodynamic selectivities but low or even opposite kinetic selectivities at low temperatures or vice versa, flexible materials might be advantageous for the design of new separation materials for such processes.

## 4. Conclusions

Based on the experimental results and the subsequent modelling, it is shown that the overall velocity of gas uptakes in flexible MOFs is predominantly determined by the rate of the structural transition. The latter is, in turn, much more strongly influenced by the applied temperature and pressure steps than conventional diffusion. Moreover, the particle size has a governing effect, with larger particles having much longer structural transition times. 

This can lead to anti-Arrhenius behavior, i.e., a slower velocity of gas uptake at higher temperatures with larger pressure steps compared to a case vice versa. The change in adsorption potential between the applied pressure step to the upper boundary of the gate-opening pressure ($p_{GOE}$) has proven to be suitable for qualitatively describing the occurrence of both Arrhenius and anti-Arrhenius behavior. Thus, the rate of structural transition is considered to be mainly governed by basic thermodynamic relations rather than by bare diffusion phenomena.

Thus, this thermodynamic basis for the overall velocities of gas uptake in flexible MOFs has a high potential for future applications as it becomes possible to systematically shift velocities of gas uptake and thus, separation selectivity in gas mixtures. This is due to the infinite number of combinations of temperature and pressure that lead to very similar uptake curves. Furthermore, if differences in host–guest interactions are to be assumed, this could further enable a fine-tuning of sorption selectivities within gas mixtures. This, however, will (have to be) be substantiated within future work. The fundamental understanding and the mathematical description of the kinetics of sorption and structural transition within flexible MOFs is an important step towards potential applications of this material class for gas separation or gas storage. 

## Figures and Tables

**Figure 1 nanomaterials-13-00601-f001:**
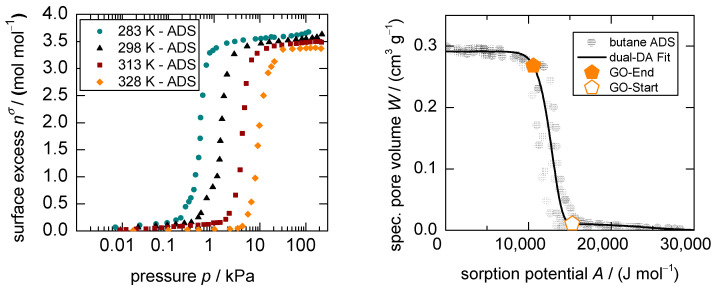
Adsorption isotherms for the system *n*-butane on **1** at 283, 298, 313, and 328 K in a logarithmic pressure scale (**left**) and the resulting Dubinin plot using the corresponding states theory for all temperatures, including the *GOS-* and *GOE*-points (**right**).

**Figure 2 nanomaterials-13-00601-f002:**
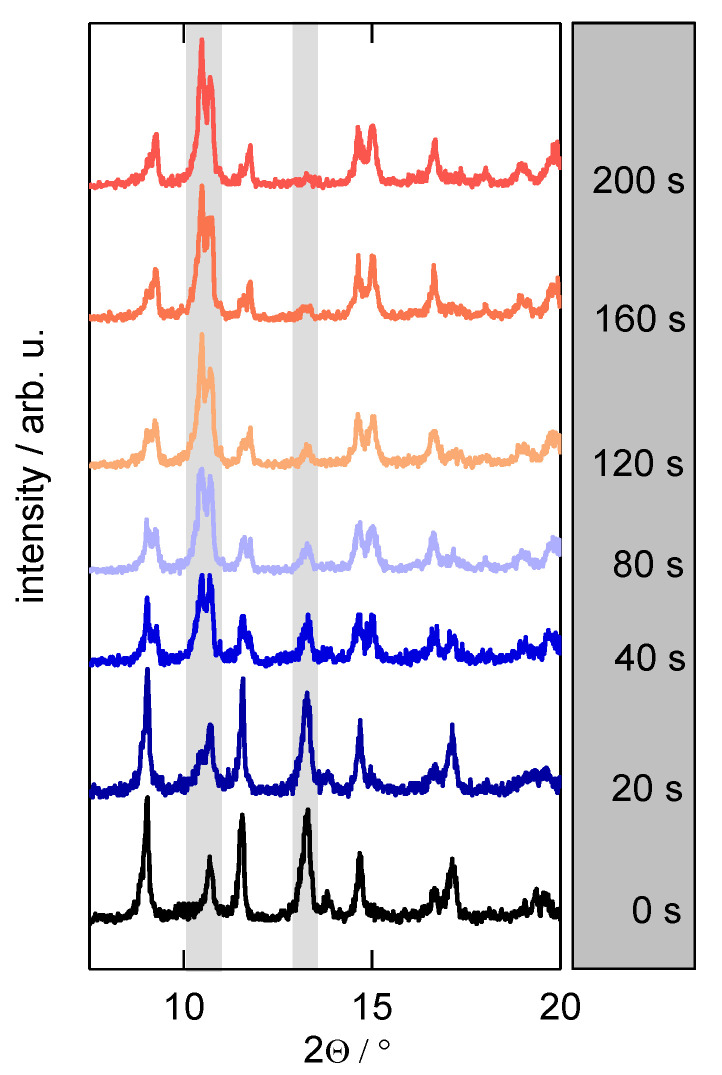
Time-dependent X-ray powder diffraction (Cu-K_α1_ radiation, stationary Dectris Pilatus detector) patterns for the uptake of *n*-butane on **1** at 298 K for a pressure step of 0 → 10 kPa in a glass capillary. The grey bars emphasize the two areas (2θ ≈ 11° and 13°) with the most significant change regarding the phase transition.

**Figure 3 nanomaterials-13-00601-f003:**
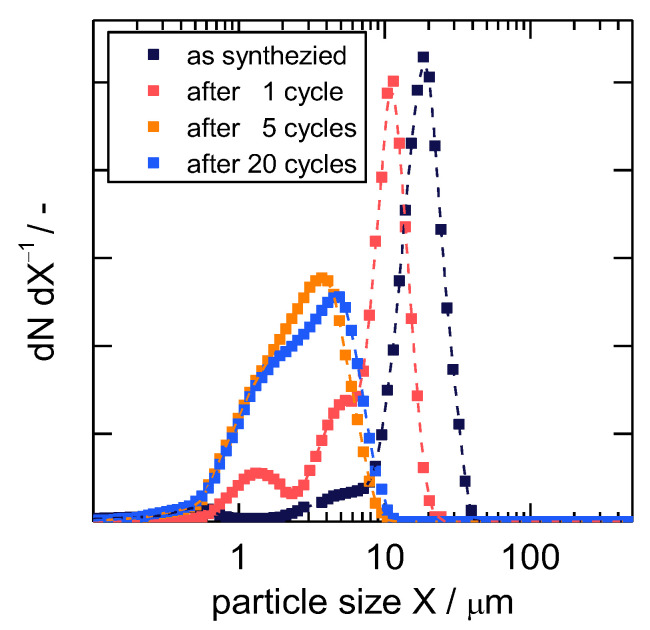
Particle size distribution for **1** after different numbers of adsorption-desorption cycles plotted as the derivative *dN*/*dX*.

**Figure 4 nanomaterials-13-00601-f004:**
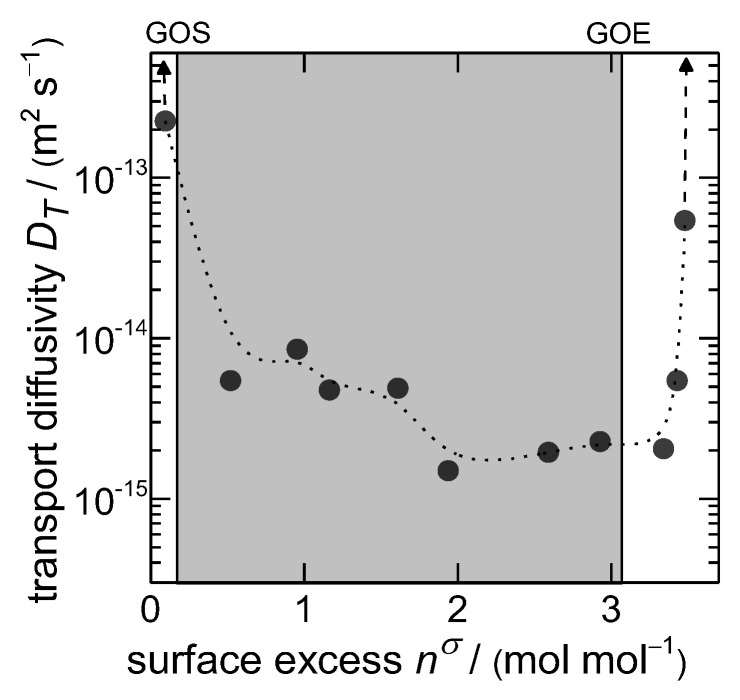
Transport diffusivities of *n*-butane on **1** in dependence of the loading. The grey area marks the loading range between gate-opening start (*GOS*) and gate-opening end (*GOE*). The arrows indicate much higher diffusivities in the regions of very low and very high pressures, which were not measurable under the experimental conditions.

**Figure 5 nanomaterials-13-00601-f005:**
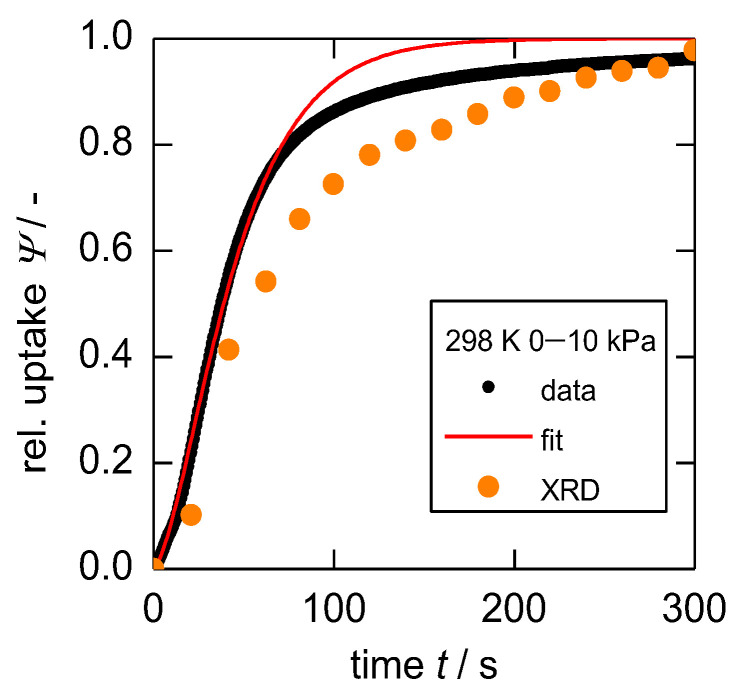
Gravimetric uptake of *n*-butane on **1** at 298 K after a pressure step 0 → 10 kPa, the fitted model presented in this work and the uptake as derived from PXRD data measured under the same conditions, representing the state of structural transition in dependence of time.

**Figure 6 nanomaterials-13-00601-f006:**
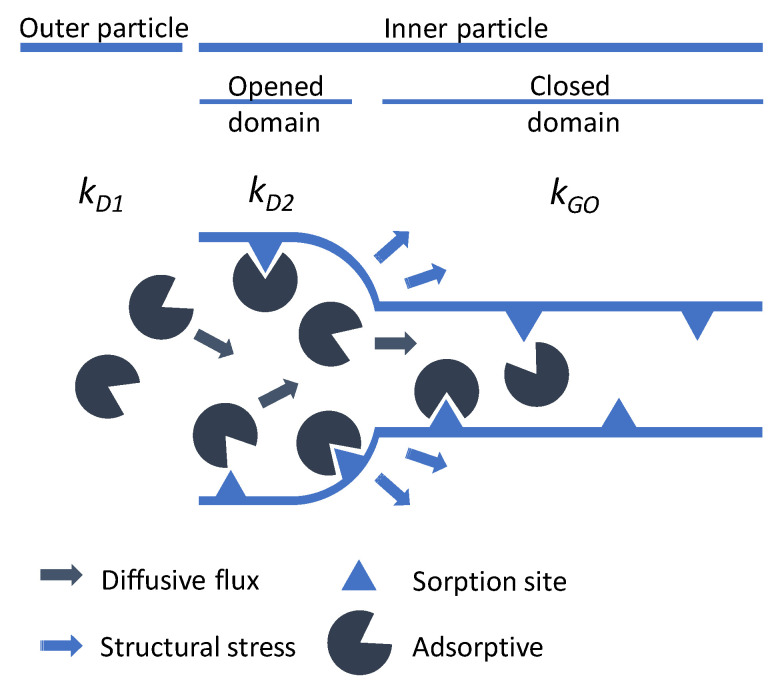
Schematic representation of the governing processes during adsorption of *n*-butane on **1** and their respective rate constants.

**Figure 7 nanomaterials-13-00601-f007:**
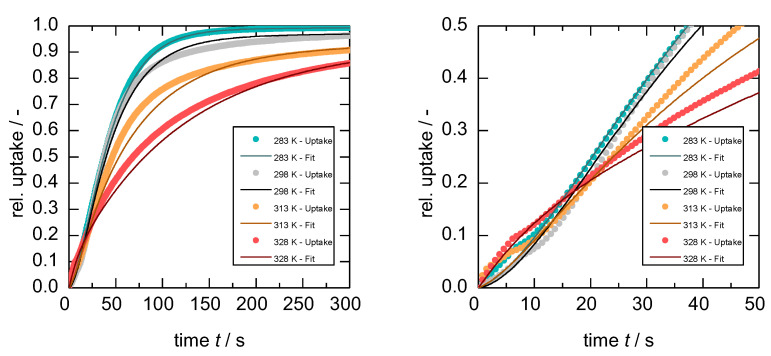
Uptake of *n*-butane on **1** at 283, 298, 313, and 328 K for a pressure step 0 → 0.04 *p*
$p_{0}^{-1}$ and their corresponding fits for the whole process (**left**) and zoomed-in for the first 50 s (**right**).

**Figure 8 nanomaterials-13-00601-f008:**
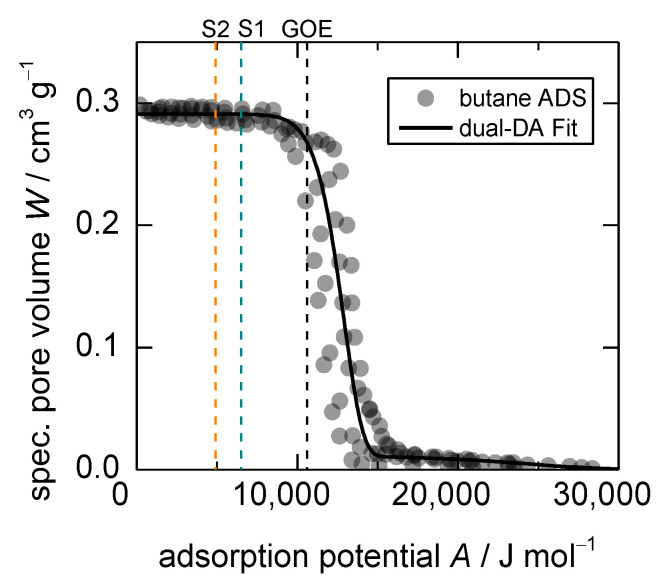
Dubinin plot of *n*-butane adsorption on **1** at 283, 298, 313, and 328 K. The position of *GOE* (*A_GOE_* = 10,600 J mol^−1^) and of the uptake equilibria for *dA_GO_* = 4100 J mol^−1^ (S1: A = 6500 J mol^−1^) and *dA_GO_* = 5700 J mol^−1^ (S2: A = 4900 J mol^−1^) with respect to *GOE* are shown as dotted lines.

**Figure 9 nanomaterials-13-00601-f009:**
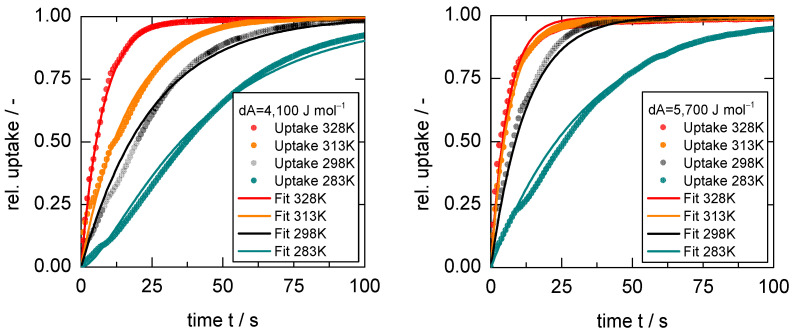
Gas uptake curves for *n*-butane on **1** at 283, 298, 313, and 328 K for changes in adsorption potential of dA = 4100 J mol^−1^ (**left**) and dA=5700 J mol^−1^ (**right**) with respect to *A_GOE_*.

**Figure 10 nanomaterials-13-00601-f010:**
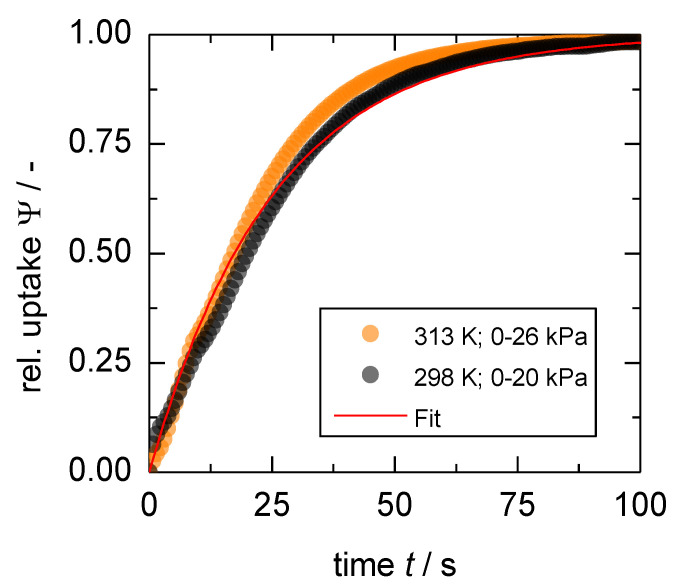
Two uptakes with the same sorption rate coefficient, although the black curve shows the relative uptake at a lower temperature and a smaller pressure step.

**Table 1 nanomaterials-13-00601-t001:** Comparison of the model fitting parameters for uptakes of *n*-butane on **1** at 283, 298, 313, and 328 K for a pressure step 0→0.04 *p*
$p_{0}^{-1}$.

*T*/*K*	$k_{D1}$/s ^−1^	$k_{D2}$/s ^−1^	$k_{GO}$/s ^−1^
283	0.03	0.03	0.070
298	0.04	0.04	0.045
313	0.08	0.08	0.016
328	0.12	0.15	0.010

## Data Availability

The datasets generated during and/or analyzed during the current study are available from the corresponding author.

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
