# Peer review of "Hydrocarbon Sorption in Flexible MOFs—Part II: Understanding Adsorption Kinetics"

_nanomaterials, 2023, doi:10.3390/nano13030601_

Round 1
Reviewer 1 Report
In this manuscript, the authors report the kinetics analysis during the n-butane adsorption by a flexible metal-organic framework. Although the structural dynamic is complex and not easily understood intuitively, the authors have done a nice job in the preliminary experimental exploration and data analysis. Impressively, the overall adsorption rate of this material is primarily depending on the step of the structural transformation, which was influenced by the temperature, pressure-step and particle size. This is a solid work and represents a new idea for the analysis of adsorption behavior of flexible adsorbents. These results should be of great interest to the community working on MOFs, gas separation, etc. I would like to recommend the acceptance of this manuscript at Nanomaterials after some minor revisions.
(1) In the introduction, the authors summarized several semi-empirical models for describing the uptake curves. However, it is very general. The applicable scope and advantage of each model should be included.
(2) I would suggest the authors to provide a schematic representation of the MOF structure and corresponding structural transformation in the supporting information.
(3) The experiment results revealed that the rate of structural transformation is mainly controlled by thermodynamic energy difference rather than kinetic diffusion. The author should explain whether this deviates from the main topic (i.e., adsorption kinetics study) of the manuscript.
(4) If possible, I would recommend that the authors include more details in the fitting of the adsorption curves.
Author Response
- Comment: In the introduction, the authors summarized several semi-empirical models for describing the uptake curves. However, it is very general. The applicable scope and advantage of each model should be included.
Response and action taken:
We thank the reviewer for his/her comments. In the updated version, we included some remarks regarding the used models. For instance, for the compressed/stretched LDF approach uses an additional exponential factor to the LDF equation, which results in improved fits, but is quite challenging to evaluate and connect to the real-world system.
- Comment: I would suggest the authors to provide a schematic representation of the MOF structure and corresponding structural transformation in the supporting information.
Response and action taken: An additional schematic representation is now included within the supporting information, showing the MOF structure in closed as well as opened stage.
- Comment: The experiment results revealed that the rate of structural transformation is mainly controlled by thermodynamic energy difference rather than kinetic diffusion. The author should explain whether this deviates from the main topic (i.e., adsorption kinetics study) of the manuscript.
Response and action taken: Indeed, the overall rate of sorption is predominantly governed by thermodynamics and not by diffusion phenomena. However, we define the uptake curves themselves as a kinetical analysis, which looks at both host guest interactions as well as diffusion. Therefore, we remain with these terms and make the definitions clearer within the abstract.
- Comment: If possible, I would recommend that the authors include more details in the fitting of the adsorption curves.
Response and action taken: As recommended by the reviewer, an additional section is now included in the experimental section 2.2.7.
Reviewer 2 Report
This is a beautiful piece of research by the Leipzig group. It describes and analysis in great detail the interplay between diffusive and structural transition effects in a flexible MOF adsorbate.
The structural changes, pore widenings, triggered by the n-butane adsorbate as a function of changes in P and T are rigorously explained by the authors from traditional (Dubinin) thermodynamics and an explanation of the (apparent) cross-over in Arrhenius behaviour is given.
The MS bears all the positive attributes of a scientific study performed by a "multi-disciplinary team" such as represented by the present authors group: rigorous experimentation followed by an in-depth thermodynamic and model-based analysis. As already anticipated by the authors in the current MS, some intriguing future work in this field to look forward to is to come from the group!
I only have very few (optional) detailed comments (that are hard to refer to given the absence of line numberings in the MS).
In the abstract the acronym LDF should be replaced by the full "Linear Driving Force" term.
In 2.2.1 the term "prohibits" needs to be raced by "avoids" or something similar.
General: I would appreciate some statements by the authors about how specific the role n-butane is in the current adsorption process and whether they expect/predict strong adsorbate effects in the triggering process.
Author Response
- Comment: In the abstract the acronym LDF should be replaced by the full "Linear Driving Force" term.
Response and action taken:
We thank the reviewer for his/her general comments on the manuscript. The aforementioned abbreviation is now introduced in the abstract to make for an easier read.
- Comment: In 2.2.1 the term "prohibits" needs to be raced by "avoids" or something similar.
Response and action taken: Thank you very much for making us aware of this mistake. Indeed, “prohibits” is too strong and is now replaced with “avoids”.
- Comment: General: I would appreciate some statements by the authors about how specific the role n-butane is in the current adsorption process and whether they expect/predict strong adsorbate effects in the triggering process.
Response and action taken: While within the discussed manuscript, only n-butane was tested as a model adsorbate, we have conducted several analyses concerning other adsorptives from the C-4 series (iso-butane, iso-butene, 1-butene) as well as other simple alkanes (methane. ethane, propane). Herein, it became evident that the adsorptive has an enormous effect on the transition rate, with iso-butene the fastest of the C-4 series and iso-butane requiring times in access of 2-3(!) weeks for pressure jumps close to the gate-opening end point. With both molecules of about the same size, this further strengthens the hypothesis that the transition itself is not driven by pure diffusion and sterical effects but rather thermodynamic relations like host-guest interactions (adsorbate effects). However, the result of this work is part of a third manuscript that we plan to publish shortly. Therefore, we would refer to this additional question within the conclusion section of this manuscript and make reference to our future work.
Reviewer 3 Report
The present paper, entitled " Hydrocarbon Sorption in Flexible MOFs – Part II: Understand-ing Adsorption Kinetics " reports an interesting work on a very important topic. Up-to-date literature was included and introduced in this article. The article is well written, summarizes a lot of valuable information in its figures and tables, and contains a lot of work.
The manuscript can be accepted for publication in the present form.
Author Response
Overall Comment: The manuscript can be accepted for publication in the present form.
Response and action taken:
No adjustments required
Reviewer 4 Report
The manuscript "Hydrocarbon Sorption in Flexible MOFs – Part II: Understanding Adsorption Kinetics" has an actual and important subject of the research field.
The work present the adsorption kinetics of n-butane on the structurally flexible metal-organic framework [Cu2(H-Me-trz-ia)2] and its complete structural transition between a narrow-pore phase and a large-pore phase. The analytical methods which supported the study were sorption gravimetry, IR spectroscopy and powder X-ray diffraction at close to ambient temperature (283, 298 and 313 K).
The presented results were interesting and useful.
The methodologies and procedures are adequate.
The reference covering the specific field of the research.
The conclusions were based on the obtained results.
The work has a very good design and argumentation.
The manuscript can be considered for publish in the presented form.
Author Response

(The authors gave the same response as above.)
